# Establishment of Stem Cell-like Cells of *Sida hermaphrodita* (L.) Rusby from Explants Containing Cambial Meristems

**DOI:** 10.3390/ijms23147644

**Published:** 2022-07-11

**Authors:** Šarlota Kaňuková, Marcela Gubišová, Lenka Klčová, Daniel Mihálik, Ján Kraic

**Affiliations:** 1Department of Biotechnology, Faculty of Natural Sciences, University of Ss. Cyril and Methodius, Námestie J. Herdu 2, 91701 Trnava, Slovakia; sarlota.kanukova@ucm.sk (Š.K.); daniel.mihalik@ucm.sk (D.M.); 2Research Institute of Plant Production, National Agricultural and Food Centre, Bratislavská cesta 122, 92168 Piešťany, Slovakia; marcela.gubisova@nppc.sk (M.G.); lenka.klcova@nppc.sk (L.K.)

**Keywords:** Virginia mallow, stem segments, cambial meristem, callus, cell biomass, stem cells

## Abstract

The in vitro cultures of plant stem cells and stem cell-like cells can be established from tissues containing meristematic cells. Chemical compounds—as well as their production potential—is among the emerging topics of plant biotechnology. We induced the callus cell biomass growth and characterized the parameters indicating the presence of stem cells or stem cell-like cells. Four types of explants (stem, petiole, leaf, root) from *Sida hermaphrodita* (L.) Rusby and various combinations of auxins and cytokinins were tested for initiation of callus, growth of sub-cultivated callus biomass, and establishment of stem cells or stem cell-like cells. Induction of callus and its growth parameters were significantly affected both by the explant type and the combination of used plant growth hormones and regulators. The responsibility for callus initiation and growth was the highest in stem-derived explants containing cambial meristematic cells. Growth parameters of callus biomass and specific characteristics of vacuoles confirmed the presence of stem cells or stem cell-like cells in sub-cultivated callus cell biomass. Establishment of in vitro stem cell or stem cell-like cell cultures in *S. hermaphrodita* can lead to the development of various applications of in vitro cultivation systems as well as alternative applications of this crop.

## 1. Introduction

*Sida hermaphrodita* (L.) Rusby (Virgina mallow) is a tall perennial herb of the Malvaceae family. It is native to southern regions of North America but is widespread also in the areas of Pennsylvania, West Virginia, Virginia, and Ohio [1]. At present, it is also grown in some European countries as an energy crop. Most research activities regarding this crop—and probably the largest growing area—are in Poland [2,3]. An increased interest in this perennial multi-purpose crop is based on the high biomass yield used for energy production, either by direct combustion or by biogas and biofuel production. The industrial use of *S. hermaphrodita* is mainly in the production of pulp and paper [4]. It can be used also in soil phytoremediation and detoxification [5,6] and has been tested as a possible substitute of other ingredients in livestock feed mixtures as well [7]. *S. hermaphrodita* contains a high amount of sugar in its secreted nectar, which allows the production of 110–315 kg of honey per hectare [8]. Pharmaceutical and medical properties and applications of *S. hermaphrodita* have not yet been published. However, many reports on plants species from the genus *Sida* L., containing approximately 200 species of herbaceous plants, were described for ethnomedicinal usage in the treatment of various diseases. Extracts from *Sida acuta*, *Sida cordifolia*, *Sida spinosa*, *Sida rhombifolia*, and *Sida veronicaefolia* are used in traditional medicine due to the content of many alkaloids, flavonoids, other phenolics, and phytoecdysteroids [9,10]. Their detailed chemical, biological, and clinical studies could lead to promising use also in modern medicine. The metabolite profiles of several members of the Malvaceae family are interesting, but the information about them is very scattered, and knowledge about *S. hermaphrodita* is almost absent. Only recently has it been proven that extract from *S. hermaphrodita* expressed effective antifungal activity against the clinical strain of *Candida albicans* and moderate activity to other *Candida* strains tested [11]. That extract was recommended for further research as a bioactive preparation and a suitable candidate for the treatment of systemic candidiasis-associated skin lesions. Secondary metabolites commonly found and characterized in other species of the genus *Sida* L.—such as apigenin with clinical effects on skin aging [12,13], and vasicin and vasinone blocking tyrosinases causing hyperpigmented skin diseases [14]—have not yet been analyzed and tested in *S. hermaphrodita*. Only information about extraction of rutin, isoquercitrin, quercimeritrin, herbacetin, coumarin, scopoletin, and scopolin in the aboveground part of the *S. hermaphrodita* were reported a long time ago [15,16]. All these compounds are known to have biological and pharmacological effects [17,18,19,20,21,22]. Even based on limited information, it can be concluded that *S. hermaphrodita* can be an interesting plant with a wide range of uses. 

The chemical compounds of *S. hermaphrodita* that can be isolated from field-grown biomass should be promising and desirable. However, isolation of target compounds directly from plant biomass has several limitations, such as differences in quality of the compounds depending on the environment, growth and harvesting conditions of plants [23], slow plant growth and synthesis of physiologically active substances and low-volume production [24], limitation to certain plant organs only, and laborious isolation of target compounds [25]. 

Alternatively, the continuous production of the desired natural plant compounds can be cultivated in the cells in vitro. This requires the development of in vitro culture systems for the cultivation of plant organs, tissue, or cells. Such production is not affected by the environment and production itself takes place in a shorter time [26,27,28]. In general, stem cells and stem cell-like cells have a crucial position in plant in vitro cultures. They prove to be the most versatile in vitro production system due to totipotency and the ability to behave as differentiated cells under specific induction conditions. Plant stem cell and stem cell-like cultures are established from tissues containing meristematic cells, and they could be used to produce cells, cell extracts, as well as specific chemical compounds, especially secondary metabolites [29,30,31]. These emerging trends and practical applications have already advanced, mainly in specific skincare cosmetic products with anti-aging, anti-inflammatory, anti-oxidative, lightening, and photo-protective effects [32,33,34,35,36]. The pluripotent plant stem cells themselves, present in stem cell niches, are maintained by complex regulatory mechanisms involving different genes, cofactors, and phytohormones [37]. All compounds involved in these mechanisms, especially phytohormones that can affect human cells and tissues, will certainly be objects of interest for research on plant stem cell biotechnology and applications in plant stem cell-based cosmetics and medical dermatology [38]. From this point of view, many if not all plant species could be of interest, especially those from which stem cells and stem cell-like cells will be isolated, in vitro multiplicated, long-term maintained, and cultivated in bioreactors in large volumes. The development of in vitro techniques also should be able to take advantage of other properties of *S. hermaphrodita*. Therefore, the aim of this study was to induce and characterize growth parameters of callus cell biomass derived from different tissues of *S. hermaphrodita* and to identify the stem cells in growing callus cell biomass.

## 2. Results

### 2.1. Callus Induction

The induction of callus was initiated from all used explants (leaves, petioles, stems, roots) obtained from aseptic seedlings. Types of explants as well as added growth hormones and regulators significantly (*p* < 0.05) influenced the time needed for induction and development of callus. The stem segments showed the earliest signs of callus formation, approximately 12 days after initiation. Other explants responded later, petiole segments after ~15 days, root segments after >17 days, and leaf segments after >20 days. After 4 weeks of cultivation, the callus formation was observed in 76.7% of the stem, 59.0% of the root, 46.0% of the petiole, and 18.4% of leaf explants, respectively. The most appropriate combinations of growth hormones and regulators were different for the callus induction and for the fresh callus weight and this was reported for each type of explant used (Appendix A). Callus biomass was heterogeneous in color and varied from white to yellow, orange, and brown. The structure of developed calli was primarily compact, only in four explants (0.1% of all) was the structure soft and watery. 

After 8 and 12 weeks of cultivation—i.e., after the first and second transfer to fresh medium—the frequency of callus formation from the four types of explants used similar to that after 4 weeks. It decreased in order: stem ˃ roots ˃ petiole ˃ leaf explants, respectively. The color was comparable to color after 4 weeks of cultivation, but the frequency of watery callus biomass slightly increased to 0.23%. Callus after 8 weeks was already formed on the entire area of the explants compared with 4 weeks, when it was only on the explant cutting surfaces.

After 16 weeks of cultivation (after the third transfers to the fresh medium) only minimal changes were observed in callus formation (Appendix A). Calli were formed with the mean frequency of 92.2% from the stem, 92.7% from the root, 69.4% from the stem, and 30.4% from leaf explants, on all combinations of growth hormones and regulators, respectively (Appendix A). Almost all (99.96%) of the calli were compact and white. Roots appeared in 25% of calli derived from root explants only.

Explants from stems, leaves, petioles, and roots were responsive for callus initiation in cultivation media supplemented with all tested combinations of growth hormones and regulators. However, high variation was observed among the growth hormones and regulators, types of explants for the time required for callus initiation, effectiveness of callus induction, as well as callus biomass growth (determined by fresh weight after 4 and 16 weeks of cultivation). Callus induction in cultivation medium without plant growth hormones or regulators was not observed.

The fresh weight of callus biomass before the first passage on the fresh callus induction medium (i.e., after 4 weeks of cultivation) was low for three types of explants, in ranges 0.024–0.128 g for petiole-derived, 0.098–0.284 g for root-derived, and 0.018–0.212 g for leaf-derived callus biomass, respectively. The weight of stem-derived callus biomass after 4 weeks was in the range 0.042–0.458 g. The most effective combination of auxin and cytokinin for frequency of callus initiation (100%) from stem-derived explants (determined at the end of the fourth week of cultivation) was 1.0 mg/L IBA + 0.5 mg/L BAP (Appendix A). The combination 1.0 mg/L 2,4-D + 0.5 mg/L KIN produced the highest callus biomass fresh weight from stem segments (Appendix A, Figure 1). The second most effective combination for callus biomass fresh weight was 1.0 mg/L NAA + 0.5 mg/L BAP.

### 2.2. Growth of Callus Biomass

The callus biomass weight is an important parameter for the multiplication of cells and cell biomass production. It was determined 4 and 16 weeks after induction of calli from primary explants. Ten different combinations of growth hormones and regulators were tested for all four explant types under the same cultivation conditions. Generally, the fresh weight of callus biomass induced from all types of explants and all combinations of growth hormones and regulators ranged after 16 weeks from 0.074 g to 8.973 g (Appendix A). Callus induction differed according to the explant type as well as the combination of growth hormones and regulators. The fresh weight of cell mass reached different values after 16 weeks of cultivation. The highest was in petiole-derived biomass on MS medium with 2,4-D + KIN, in root-derived biomass on MS with NAA + BAP and 2,4-D + BAP, and in leaf-derived biomass on MS with IBA + BAP and 2,4-D + KIN, respectively (Appendix A). 

Even after 16 weeks of cultivation, the fresh weight of callus cell biomass reached the highest values in stem-derived biomass, especially in media with combinations of 1.0 mg/L IBA + 0.5 mg/L BAP (8.973 g) and 1.0 mg/L 2,4-D + 0.5 mg/L KIN (7.244 g), respectively (Appendix A, Figure 2). These two media seem to be the most appropriate if massive production of callus-derived biomass is needed permanently. 

Callus biomass on these media was massive, looked very fresh, and capable to further maintaining, multiplication, and production of cells (Figure 3). The most notable was the increase in fresh weight of callus cell biomass between the 4th and 16th weeks of cultivation on the medium with IBA+BAP (Figure 2, Appendix A).

### 2.3. Stem Cell-like Cells

Callus biomass developed from stem segments was induced the most efficiently and the fresh weight after 4 as well as 16 weeks of cultivation was also the highest. The presence of innately cambial meristematic cells in stem-derived explants suggests that cells with properties of stem or stem cell-like cells should be present also in the growing multicellular callus biomass along with dedifferentiated and differentiated cells. This could be the basis for the establishment of the stem cell-like cell culture in vitro. The presence of stem cell-like cells in growing cell biomass should be supported by the morphological features of cells developed on the callus surface. Those cells were white to transparent, compact (Figure 3), and should also be small with a thin primary cell wall [39]. 

Another physical parameter of the stem cell-like cells was the structure of vacuoles after staining with the neutral red. Abundant small spheric vacuoles or vacuole-like structures could be observed in cambial meristematic cells, while usually only one large vacuole could be observed in dedifferentiated cells [29]. Microscopic observation confirmed the presence of such vacuolar differences. The cells with highly abundant small vacuoles should be the stem cell-like cells, in comparison to dedifferentiated callus cells with typically only one large vacuole (Figure 4).

The callus cell biomass containing stem cell-like cells also matched typical growth parameters [40]. The volume and related fresh weight of stem-derived callus cell biomass that originally contained cambial meristematic cells were after four cycles (16 weeks) of sub-culture on solid medium was significantly higher than in cell biomass derived from leaves, petioles, and roots, respectively (Figure 5). This was most pronounced on media with 1.0 mg/L IBA + 0.5 mg/L BAP and 1.0 mg/L 2,4-D + 0.5 mg/L KIN. The biomass increment in these two variants was 3.39 × 10^3^% and 1.48 × 10^3^%, respectively. Even greater differences in growth parameters after 22 months of cultivation were characteristic between cambial meristematic cells and dedifferentiated cells also in *Taxus cuspidate* [29]. Growth parameters of callus biomass initiated from leaf, stem, and root explants were almost identical, with only a slight increase in cell biomass weight. An increase in callus cell biomass derived from the stem explants containing cambial meristem cells was significant, even after the third passage (12 weeks). Moreover, stem-derived callus cell biomass maintained a uniform and vigorous growth status.

## 3. Discussion

Seeds of *S. hermaphrodita* must be scarified before germination because their germina-tion without this step is in the range of only 5–10% [41] due to strong physical dormancy [42]. Scarification by concentrated sulfuric acid increased seed germination in our experiments, from 8.7% to 51.4%. *S. hermaphrodita* can also be reproduced asexually by rhizomes; however, the commonly used approach in Europe is the planting of seedlings [5]. Seedlings can be effectively produced by micropropagation techniques in vitro as have been developed for other energy crops [43,44,45,46]. A simpler approach is the indirect micropropagation by plant regeneration from morphogenetically competent cells in callus tissue. However, such a procedure has not been developed or published for *S. hermaphrodita*. The possibility of developing shoots and roots by indirect organogenesis from calli derived from leaf discs and leaf stems has been reported very briefly [47]. A protocol for in vitro multiplication of related species *Sida cordifolia* L. has been published [48]. Initiation of callus cultures is an intermediate stage for in vitro micropropagation, and stem cells retained in the tissues used as explants contribute to callus formation under certain environmental conditions [49], especially in the presence of auxins and cytokinins stimulating the stem cells remaining in excised explants to division and inducing the adjacent potential of cells to form callus in vitro [50]. Various combinations of auxins + cytokinins applied in the ratio 1.0 mg/L + 0.5 mg/L for induction and growth of a callus are common for different explants in many plant species. As the most effective for long-term callus growth derived from stem segments of *S. hermaphrodita* was combination IBA + BAP in these concentrations. The combination of 2,4-D + KIN also effectively produced callus biomass. Conversely, some combinations (IBA + KIN, IBA + 2iP) and 2,4-D alone were not very effective (Figure 2).

The presence of stem cells or stem cell-like cells, observed in this study in the growing callus biomass of *S. hermaphrodita*, is significant due to their role in callus formation, callus growth, and significant association with the cell dedifferentiation process [49] by which cells enter the stage with increased developmental potency [51]. However, this study was not focused on plantlet regeneration and micropropagation, but on the long-term cultivation and growth of callus cell biomass and formation of stem cells or stem cell-like cells. Callus cultures themselves offer a wide range of biotechnological applications [52,53] aimed at the highly desirable application of in vitro cultures, the production of various natural products [31,53,54], especially secondary metabolites [28]. They must be maintained for a long time with the continuous growth of cell biomass and production of required metabolites. Callus initiation and growth usually occur regardless of plant species and explant type, but this process also requires testing of various explants, nutrient media, culture conditions, and especially growth hormones and regulators. From the point of view of *S. hermaphrodita*, the stem-derived explants were very effective in callus initiation as well as callus biomass growth during 4 months of cultivation. Weight of callus cell biomass derived from stem segments was 11–15 times higher in comparison with leaf, petiole, and root explants after 16 weeks of growth (Figure 5, Appendix A). The establishment of cell suspension cultures from callus cultures typically has a higher potential for the production of cell biomass and secondary metabolites, mainly of pharmaceutical and medicinal interest [28,55]. Similar applications are associated at present with in vitro cultured plant stem cells. Currently, the most promising field of plant stem cells is cosmetic manufacture due to the antioxidant and anti-inflammatory properties of stem cell derivatives [36,56]. Plant stem cells can be applied as active stem cells or stem cell extracts, especially in skin care products [32,57]. Another interest for the stem cells is production of secondary metabolites in vitro, such as paclitaxel [29], ginkgolides [58], ginsenoside [59], and others. Although such studies in *S. hermaphrodita* have not yet been reported, earlier studies revealed such potential of cells from the vegetative tissues and organs of *S. hermaphrodita* [15,16]. The establishment, maintenance, and propagation of stem cell lines in vitro would greatly expand their use.

The in vitro culture of plant stem cells can be established from post-embryonic stem cell systems, shoot meristems, root meristems, and lateral meristems, respectively [60]. The lateral meristems are usually the most readily available as they are located between the phloem and xylem or near the surface layer of the root and stem. They are related to procambium and cambium, also called as meristems of the conductive system. These cells are clonogenic precursors whose daughter cells can either remain stem cells or undergo differentiation [61]. Plant stem cell cultures were established from innately undifferentiated lines of cells derived from isolated cambial meristematic cells after peeling off cambium, phloem, cortex, and epidermal cells from the xylem [29,62,63]. Another way was scraping of new cells that appeared in the in vitro cultivated vascular cambium explants [64]. The amount of cambial meristematic cells has been increased simply by pre-treatment of explants with indole-3-butyric acid (IBA) before cultivation on solid medium [40]. Stem cell cultures established without these techniques, from already established cell suspension cultures derived from calli, were used in *Lycopersicon esculentum* [57]. Structural and growth characteristics of callus biomass derived from stem segments of *S. hermaphrodita* predicted the presence of innate cambial stem cells. Callus is predominantly formed from a pre-existing population of stem cells under certain environmental conditions [65,66]. Innate stem cells present in cultivated tissue may communicate with neighboring cells by releasing specific signals during dedifferentiation of neighboring cells [49]. Besides division and generation of cells that ultimately undergo specialization or differentiation, they also give rise to new stem cells [61]. This was confirmed by the ultrastructural parameters of the cells after staining with the neutral red accumulating in plant vacuoles [67]. Stem cells or stem cell-like cells contained abundant and small vacuoles, whereas dedifferentiated cells were characterized by the presence of one large vacuole. These characteristics were consistent with cambial stem cells presented in previous studies on various plant species [29,62,63]. Even though the stem cells or stem cell-like cells of *S. hermaphrodita* were not isolated from separated layers containing cambium or by scraping technique, they were identified in cultured callus cell biomass. Thus, the present study offers another possible and very simple approach based on callus biomass initiated, maintained, and propagated on a solid medium.

## 4. Materials and Methods

### 4.1. Seeds Germination In Vitro

Seeds of the *Sida hermaphrodita* (L.) Rusby were harvested from a long-term field experiment maintained in the Research Institute of Plant Production (Piešťany, Slovakia). Seeds were scarified before germination for 20 min in concentrated sulfuric acid. After rinsing under tap water, the seeds were air-dried, surface sterilized in 80% ethanol for 30 s, in 5% sodium hypochlorite solution for 10 min, and washed four times in sterile water. Scarified and surface sterilized seeds they were placed on the basal half-strength Murashige and Skoog (½MS) medium [68] containing half concentrations of inorganic salts, sucrose (3%, *w*/*v*), and agar (0.8%, *w*/*v*), pH 5.7–5.8. Germinated seeds and seedlings were cultivated at 23 ± 2 °C under a photoperiod of 16 h light/8 h darkness.

### 4.2. Plant Explants and Callus Induction

Four-week-old in vitro seedlings were taken for initiation of callus tissue. Explants were excised from leaves, petioles, internodes, and roots. The leaf explants were approximately 5 × 5 mm in size and the petiole, stem, and root explants were about 5 mm in length. Twenty to 25 explants of each type were used for initiation of a callus on basal MS supplemented by sucrose (3%, *w*/*v*), agar (0.8%, *w*/*v*), pH 5.7–5.8, 1 mg/L 2,4-dichlorophenoxyacetic acid (2,4-D) alone or combinations of 1 mg/L of auxin + 0.5 mg/L of cytokinin (Appendix A). The auxins used were 2,4-D, 1-naphthylacetic acid (NAA), or indole-3-butyric acid (IBA), used the cytokinins used were 6-(γ,γ-dimethylallylamino)purine (2iP), kinetin (KIN), or 6-benzylaminopurine (BAP). Explants were cultivated at 23 ± 2 °C in the dark and sub-cultured at 28-day intervals on the same cultivation media. Cultivation medium, growth hormones and regulators, and all other chemicals were from Duchefa Biochemie B.V. (Haarlem, The Netherlands). 

Calli were evaluated for texture, color, time to callus emergence, callus induction frequency (% of cultivated explants producing callus), fresh weight, and the appearance of regenerated roots (Appendix A).

### 4.3. Stem Cell-like Cells Analysis

Microscopic analysis of cells was performed in friable callus tissue by dyeing of vacuoles using the modified method of Lee et al. (2010). Cells taken from callus biomass were stained with 0.01% (*w*/*v*) neutral red (3-amino-7-dimethyl-amino-2-methylphenazine hydrochloride, Merck KGaA, Darmstadt, Germany) for 10 min and washed with 0.1 M phosphate buffer, pH 7.2. Samples were prepared by the squash smear technique and observed using the Leica DM6000 Upright Optical Microscope (Leica Microsystems GmbH, Wetzlar, Germany).

### 4.4. Statistical Analysis

To obtain the normal distribution of the data, the Box–Cox transformation was used using the PAST (PAleontological STatistics) software version 3.19 [69]. Obtained data were evaluated by the analysis of variance (ANOVA) using the Statgraphics software version 19.2.01 (Statgraphics Technologies, Inc., The Plains, VA, USA). Significant differences were compared using the least significant difference (LSD) test at 5% level of significance (*p* < 0.05). All experiments were carried out in five replications for each treatment and each replication contained four to five explants.

## 5. Conclusions

The effect of explant type and plant growth hormones and regulators on induction of a callus and establishment of callus biomass containing stem cells or stem cell-like cells of *Sida hermaphrodita* (L.) Rusby was the subjects of this study. Stem-derived explants were selected as the most suitable for callus initiation, continual callus growth, callus cell biomass production, and establishment of stem cells or stem like-cells. The most effective combination of the growth hormone and regulator was 1.0 mg/L IBA + 0.5 mg/L BAP. Explants derived from roots, petioles, and leaves formed calli with much lower efficiency. Morphological and growth parameters of stem-derived callus biomass were different from calli initiated from other explants. Those parameters indicated the presence of stem cells or stem cell-like cells in growing callus biomass. The presence of a large proportion of stem cells or stem-like cells with characteristic vacuole structure was confirmed in the callus biomass that intensely increased fresh weight during sub-cultivation cycles. The establishment of stem cells and stem-like cells in long-term cultivated callus biomass was reported for the first time in *S. hermaphrodita*. 

## Figures and Tables

**Figure 1 ijms-23-07644-f001:**
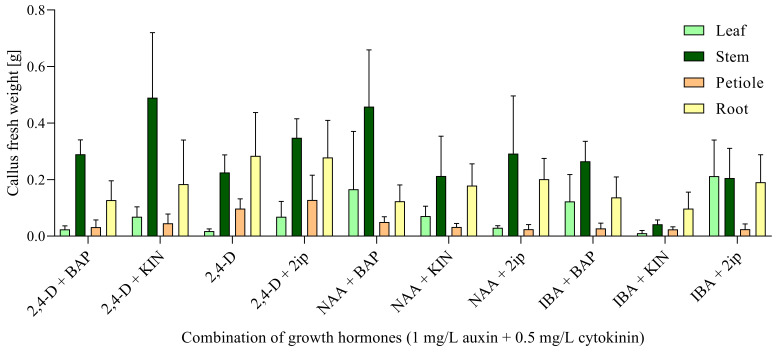
Induction of callus after 4 weeks of cultivation on media with different combinations of growth hormones and regulators. Data are presented as means ± SDs (*n* = 5).

**Figure 2 ijms-23-07644-f002:**
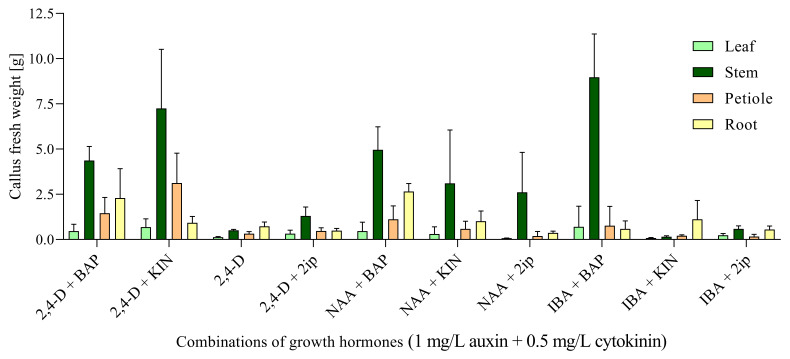
Production of callus cell biomass after 16 weeks of cultivation on media with different combinations of growth hormones and regulators. Data are presented as means ± SDs (*n* = 5).

**Figure 3 ijms-23-07644-f003:**
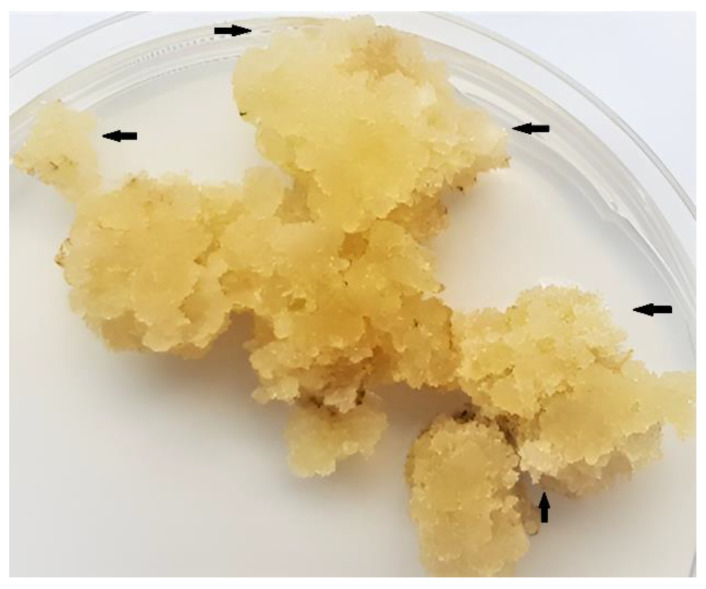
Appearance and structure of callus cell biomass of *S. hermaphrodita* grown on MS medium with 1.0 mg/L IBA + 0.5 mg/L BAP after 16 weeks. Black arrows indicate areas at the callus surface with areas of whiter cells.

**Figure 4 ijms-23-07644-f004:**
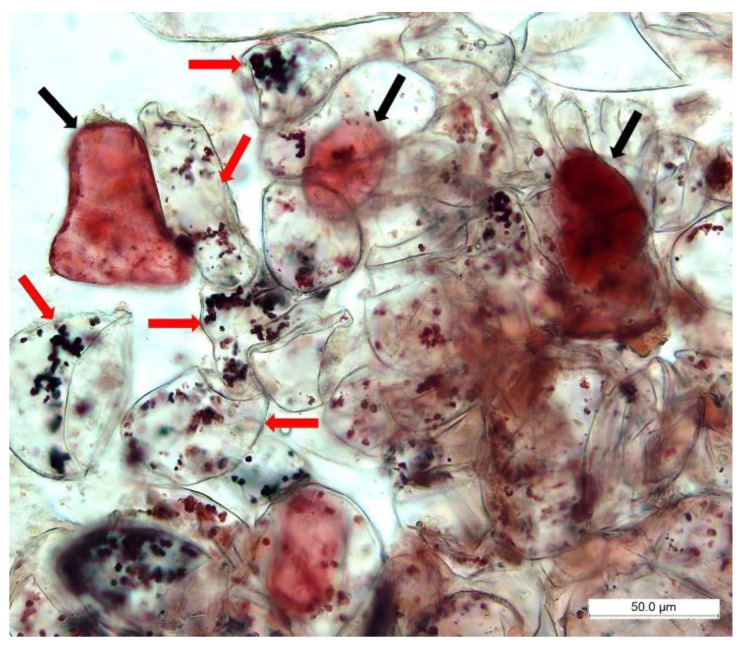
Stem cell-like cells and dedifferentiated cells of *S. hermaphrodita* in callus cell biomass. Red arrows—stem cell-like cells abundant in small colored vacuoles; black arrows—dedifferentiated cells with only one large vacuole.

**Figure 5 ijms-23-07644-f005:**
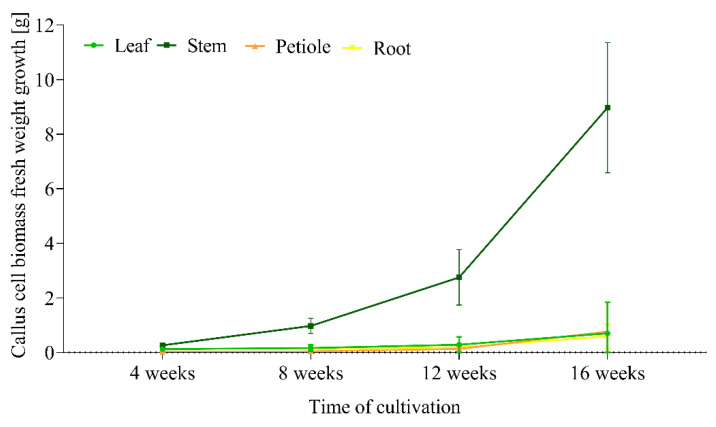
Growth dynamics of callus biomass derived from four explant types during 16 weeks of cultivation on solid MS medium with 1.0 mg/L IBA + 0.5 mg/L BAP.

## Data Availability

Not applicable.

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
