# Peer review of "Establishment of Stem Cell-like Cells of Sida hermaphrodita (L.) Rusby from Explants Containing Cambial Meristems"

_ijms, 2022, doi:10.3390/ijms23147644_

Round 1
Reviewer 1 Report
Dear authors,
Hope you are doing well. I read your manuscript and I think that it is very interesting. Also, it's generally well written. I give you my comments:
Major:
- Abstract. I think that one or two brief introduction sentences should be included in the abstract. You should make clear what the research aim is.
- Discussion. Although I think that the discussion is well written and covers all research I think that it can be improved. From my point of view, the discussion sometimes remind me an introduction. For instance, it is important to highlight that certain auxin-cytokinin combinations works better than others. The same for the explant type. Perhaps, it could be interesting to include some figure cites to lead the discussion.
Minor:
-Lines 131-132. Do you refer to stem explants? Please, include it.
-Line 307. "with". I think that this sentence It would be better understood if you deleted that word.
- Line 308. Used?
-Section 4.3 in methodology. How were the cuts made? by hand-made or microtome? squash?
Reviewer 2 Report
1. The subject “Establishment of stem cell-like cells from cambial meristems containing explant of Sida hermaphrodita (L.) Rusby” change into “The Induction callus of Sida hermaphrodita (L.) Rusby.
2. In content,to use “Plant growth regulators,PGRs” substitute “Growth hormone “.
3. To explain”How to identify the stem cells in growing callus cell biomass?” The organogenesis will be from compact callus and mass propagation could be from watery callus by suspension subculture.
4. The color of callus (Fig.3) looks like orange not as describe in content (white).
5. The order of references follow as published year is better,and the position of published year should be uniformity. For example,the published year follow as author’s name not at last (no.63).
Reviewer 3 Report
The manuscript is written in a nice, simple and understandable manner. It is a pleasure to read. There are few minor spelling or language mistakes (corrections are proposed in the text).
However, there is an improper description or interpretation of the results. The Authors describe data which is not supported with the data. If the marginal means are not analyzed statistically, the general information concerning the effect of the explant type or PGRs separately cannot be given. The results should be rewritten or additional data given.
The other concern is the discussion. There is no comparison of the effect of the PGRs on sida callus growth in tissue culture. A large amount of information given is appropriate rather for the introduction part. Try to add some relevant information for this part.
I cannot upload more than one file, although there are changes proposed for the table as well. The 'colour' should be 'color' and the title should have the information added in the end ' in tissue culture'.
More remarks are given in the manuscript.

Round 2
Reviewer 1 Report
Dear authors,
Thank you for your comments and changes. From my point of view, now I think that it is ok.
Best regards,
Reviewer 3 Report
Thank you for the answers to my questions and remarks as well as for changing the problematic matters in the manuscript. I recommend to accept it, additionally my answers are given in the text uploaded.